behaviour/ecology/physiology

body temperature, hyperthermia, parental care, thermal constraint

**Author for correspondence:**
Simon Tapper
e-mail: simontapper@trentu.ca

# Experimental evidence that hyperthermia limits offspring provisioning in a temperate-breeding bird

Simon Tapper[1], Joseph J. Nocera[3] and Gary Burness[2]

[1]Environmental and Life Sciences Graduate Program, and [2]Department of Biology, Trent University, 1600 West Bank Drive, Peterborough, Ontario, Canada
[3]Faculty of Forestry and Environmental Management, University of New Brunswick, 28 Dineen Drive, Fredericton, New Brunswick, Canada

ST, 0000-0003-3237-291X

In many vertebrates, parental care can require long bouts of daily exercise that can span several weeks. Exercise, especially in the heat, raises body temperature, and can lead to hyperthermia. Typical strategies for regulating body temperature during endurance exercise include modifying performance to avoid hyperthermia (anticipatory regulation) and allowing body temperature to rise above normothermic levels for brief periods of time (facultative hyperthermia). Facultative hyperthermia is commonly employed by desert birds to economize on water, but this strategy may also be important for chick-rearing birds to avoid reducing offspring provisioning when thermoregulatory demands are high. In this study, we tested how chick-rearing birds balance their own body temperature against the need to provision dependent offspring. We experimentally increased the heat dissipation capacity of breeding female tree swallows (*Tachycineta bicolor*) by trimming their ventral feathers and remotely monitored provisioning rates, body temperature and the probability of hyperthermia. Birds with an experimentally increased capacity to dissipate heat (i.e. trimmed treatment) maintained higher feeding rates than controls at high ambient temperatures (greater than or equal to 25°C), while maintaining lower body temperatures. However, at the highest temperatures (greater than or equal to 25°C), trimmed individuals became hyperthermic. These results provide evidence that chick-rearing tree swallows use both anticipatory regulation and facultative hyperthermia during endurance performance. With rising global temperatures, individuals may need to increase their frequency of facultative hyperthermia to maintain nestling provisioning, and thereby maximize reproductive success.

# 1. Introduction

The impact of environmental heat exposure on physical performance is well documented in humans (reviewed in [1]), and to a lesser extent, other endotherms ([2], briefly reviewed in [3]).

In general, increased heat exposure contributes to the development of thermoregulatory strain and weakened performance (known as hyperthermia-induced fatigue) [3]. For example, in marathon runners, an ambient temperature increase from 10 to 22°C causes competitors to slow anywhere from 2 to 10%, dependent upon on an individual's level of conditioning [4]. In non-human endotherms, experimental exposure to high ambient temperatures reduces flight time in birds [5,6] and running duration in mammals [7,8]. These declines in performance are presumed to be evolutionarily adaptive, preventing dangerously high increases in body temperature ($T_b$) that occur from increases in metabolic production during exercise [9].

The mechanistic link between rising $T_b$ and reduced physical performance is not well understood, but it is thought to be a dynamic process involving a number of sensory cues (e.g. partial pressure of respiratory gases, blood pH, neurotransmitter production, skin temperature) [10]. Together, these cues result in progressive inhibition of the brain areas responsible for motor activation [3], prior to an individual reaching a dangerously high $T_b$ [10–12]. There are several lines of evidence, both correlative and experimental, supporting such 'anticipatory regulation' of $T_b$. In some species, for instance, running performance progressively falls as hypothalamic temperature increases, and heating the hypothalamus experimentally produces the same effect [13,14]. In humans, the behavioural strategy of pacing during exercise is evidence for anticipatory regulation, because individuals modify their performance before physiological impairment occurs [15–17]. When exercising in the heat, for instance, human athletes allowed to self-pace maintain a similar $T_b$, despite differences in performance [11,16]. Recent evidence suggests that free-ranging animals may use a similar mechanism of pacing to regulate body temperature. For instance, migrating common eiders (*Somateria mollissima*) stop their flying bouts prior to reaching high $T_b$, and stoppage is better explained by the accumulation of body heat, rather than a need to feed [18]. In addition to pacing, some avian species appear to simply mitigate the potential for hyperthermia by avoiding high temperatures during flight [19].

In contrast to anticipatory regulation, animals may, however, allow their $T_b$ to rise in the short term by passively storing heat in body tissues. This strategy of facultative hyperthermia is employed by a number of species [2,20–23] and in different contexts. For instance, both prey and predator species may store a large percentage of metabolic heat produced during running [21], which may allow for increased hunting success or survival. In birds, facultative hyperthermia is commonly reported among desert species [24], and is an important adaptation for retaining water, as stored heat can be dissipated by non-evaporative mean when ambient temperature ($T_a$) is lower [25]. Although facultative hyperthermia is commonly reported among desert birds, it could presumably be employed by non-desert avian species during periods requiring high thermoregulatory demands, such as during running and flying [5,6,26].

Parental care is often considered to be an energetically demanding activity, involving raising offspring over several weeks, at sustained metabolic rates that are thought to border an energetic ceiling [27]. Chick-rearing birds, therefore, may opt to employ facultative hyperthermia as a means to maintain a consistent offspring feeding rate, even when environmental heat loads are high. To understand how animals may regulate sustained activity, we remotely monitored the daytime $T_b$ of free-ranging tree swallows (*Tachycineta bicolor*) over a two-week period during the breeding season, when individuals were feeding their young. We experimentally trimmed the ventral feathers of females to increase their heat dissipation capacity (as has been demonstrated in [28]) and asked: (i) do provisioning swallows manage their $T_b$ using mechanisms consistent with anticipatory regulation? and (ii) do provisioning swallows employ facultative hyperthermia? We recognize that our predictions are not mutually exclusive, and in fact individuals may employ a combination of strategies depending on time of day or environmental conditions (e.g. humid and warm versus humid and cool). Nonetheless, we predicted that if anticipatory regulation is a mechanism by which swallows regulate $T_b$, then trimmed birds will be capable of maintaining higher provisioning rates [29], but they will do so with the same average $T_b$ as control birds. If birds use facultative hyperthermia during nestling provisioning to avoid reducing daytime activity, then we expected no differences in provisioning rates between the two treatments, but controls would have on average higher $T_b$ compared with trimmed birds and would spend proportionally more time hyperthermic.

# 2. Material and methods

## 2.1. Study area and species

We conducted this study in May–July 2018, using two nest-box breeding populations of tree swallows located at the Trent University Nature Areas, Peterborough, Ontario, Canada (44°21′ N, 78°17′ W) and at the Lakefield Sewage Lagoon, Lakefield, Ontario, Canada (44°24′58.3″ N, 78°15′26.8″ W). Nest-boxes were largely exposed to direct sunlight at the Sewage Lagoon, and either shaded or in direct sunlight at the Nature Areas. The two sites are approximately 10 km apart. Females at both sites typically lay clutches of five to seven eggs, with one egg laid each day. Birds at the Sewage Lagoon begin laying one to two weeks before birds at the Trent Nature Areas. Once a clutch is completed, females incubate the eggs for approximately 14 days. Nestlings typically hatch synchronously and fledge 18–22 days post-hatch [30].

## 2.2. Field methods

Beginning in May 2018, we checked nest-boxes every other day until the presence of nest material was discovered, at which point we monitored the boxes every day until clutch completion. Using a non-toxic marker pen, we numbered eggs sequentially as they were laid, and we considered clutches to be complete when two consecutive days passed without the appearance of a new egg. The date the last egg was laid was considered to be day 0 of incubation for a given clutch. Hatch date for the brood (day 0 of chick rearing) was considered as the first day when nestlings hatched. In our population sample, hatch dates ranged from 28 May to 27 June 2018.

To quantify $T_b$ and provisioning rate remotely, we captured female tree swallows during late incubation (range = day 7–10 post-clutch completion). Upon capture females were aged (second-year (SY) or after-second-year (ASY)) based on plumage coloration [31], and implanted with a thermal-sensitive passive integrated transponder (PIT) tag (accuracy ± 0.5°C, Bio-Therm13, Biomark, Boise, ID, USA) subcutaneously into the nape of the neck, following McCafferty et al. [32]. Our metric of $T_b$ was, therefore, subcutaneous $T_b$, as measured via PIT tags. In small birds, subcutaneous $T_b$ is an excellent indicator of deep/core $T_b$ [32–34] and we, therefore, assumed that our metric of hyperthermia would be accurate relative to a metric calculated with deep $T_b$. It should also be noted that PIT tags can shift positions post-implantation, but if such shifts did occur in our study, it should affect both treatments similarly, and would, therefore, not impact our conclusions. The PIT tags were read by Biomark HPR Plus readers which we connected to a loop antenna (17.5 cm) and positioned so they encircled the nest-box entrance. We cycled three readers among nests daily, so that each nest received a reader for approximately 24 h three times throughout chick-rearing (early, middle and late-stage provisioning). Early, middle and late-stage provisioning were defined as days 2–5, 6–9 and 10–14 post-hatch, respectively. We set the delay interval (i.e. the amount of time required between successive reads of the same tag) to 10 s, to maximize precision while avoiding a large number of reads that would occur during continuous recording.

To increase the rate of heat loss in female tree swallows, we recaptured individuals during early nestling provisioning (range: day 1–2 post-hatch) and performed the experimental manipulation (experimental trimming versus handling). Experimental manipulations were randomly scattered among boxes and the two study sites. Upon capture, we randomly assigned females to either a trimmed or control treatment based on a coin flip. In the trimmed treatment, we used scissors to trim the contour and downy feathers covering the brood patch to expose the bare skin underneath, following Tapper et al. [29]. Birds in the control condition were handled but released with their feathers intact. The area of feathers removed represented approximately 7% of the surface area of the bird, which has been shown to impact provisioning performance in this study population [29]. Treatments were approximately balanced across age ($n_{control\ SY} = 2$, $n_{control\ ASY} = 6$, $n_{trimmed\ SY} = 4$, $n_{trimmed\ ASY} = 4$), timing of breeding (mean lay date ± 1 s.d.: $n_{control} = 8.4 \pm 4.8$, $n_{trimmed} = 9.9 \pm 6.3$) and clutch size (mean clutch size ± 1 s.d.; $n_{control} = 5.5 \pm 0.8$, $n_{trimmed} = 4.5 \pm 0.9$), where s.d. is the standard deviation.

## 2.3. Data compilation and organization

For data compilation and statistical analyses, we used R v. 4.0 [35]. Nearly all adult females that were assigned a treatment were caught when nestlings were 1 day old (15/16 females); one female was

caught on nestling day 2. To minimize the effects of capture bias on subsequent behaviour, we only included data collected between nestling ages 3 and 14 days. The sample size for all analyses includes 16 birds ($n_{control} = 8$, $n_{trimmed} = 8$), unless otherwise stated. For all analyses, we included data from 05.00 to 21.00 h, because swallows are relatively inactive between 21.00 and 05.00 h and because most birds started to show decreases in $T_b$ starting at approximately 20.30 h.

### 2.3.1. $T_b$ and feeding rate

We calculated the average hourly $T_b$ (°C) for each bird, obtaining multiple measurements per individual across several days (the mean number of days per individual: control = 5.75 days and trimmed = 6.25 days; mean ± 1 s.d. number of data points per bird = 54 ± 35). Before doing so, we first looked for any aberrant $T_b$ measurements by plotting the raw data and the hourly averages for each bird. We found that $T_b$ was abnormally low (approx. 2°C) between 05.00 and 13.00 h for one bird on one day relative to its $T_b$ between 05.00 and 13.00 h on all other days (i.e. 13% of all observations for that bird), and so we removed these data from all analyses. Previously, we reported nestling provisioning rate derived from using a combination of temperature- and non-temperature-sensitive PIT tags [29]. Here, we use a subset of those data (i.e. temperature-sensitive PIT-tagged birds only) as a dependent variable, which allows us to make direct comparisons with maternal body temperature. Details on how we calculated and analysed feeding rate are described in the electronic supplementary material.

### 2.3.2. Defining normothermia and hyperthermia

To estimate normothermia, we first required each individual's resting, or inactive, modal $T_b$, which we calculated using the individual's $T_b$ during the active phase of the day (i.e. 05.00–21.00 h), following an approach similar to Smit et al. [36]. Because we were unable to know whether a bird was active or inactive when away from the nest and out of view, we considered a bird to be 'resting' when it remained at the nest-box for greater than or equal to 5 min, which was 6.2% of all nest-box visits (527 visits/8431 visits) ($n_{control} = 8$, $n_{trimmed} = 8$). We chose our cut-off time as greater than or equal to 5 min because we assumed this would be enough time for a bird to cool down and provided sufficient data to calculate a resting modal $T_b$ for each individual. Our definition of 'resting' also included observations of brooding females, and while we recognize this could lead us to overestimate normothermic $T_b$ (via increases in maternal metabolic rate and subsequently $T_b$, [37]), this should have little impact on our conclusions because it affected both treatments equally. After calculating the modal $T_b$ per bird at rest, we then calculated a heterothermy index (HI) [38,39] to estimate the variance around resting normothermic $T_b$ (i.e. greater than or equal to 5 min, 05.00–21.00 h, between nestling ages 3 and 14 days). The HI is similar to standard deviation, but measures the deviation away from modal $T_b$, and is calculated using the following formula:

$$HI = \frac{\sqrt{\sum (T_{mod} - T_{b,i})^2}}{n - 1},$$

where $T_{mod}$ is the modal $T_b$ of an individual, $T_{b,i}$ is the $T_b$ at time $i$ and $n$ is the number of times $T_b$ is sampled (in our case, this differs per bird).

We defined an individual as hyperthermic when its $T_b$ surpassed its modal $T_b$ + HI (i.e. upper bound on $T_b$ during 'resting'). We, therefore, categorized each $T_b$ observation (i.e. every instance in which the bird was logged on the reader) as normothermic or hyperthermic and subsequently calculated the total number of observations for which each individual was present at the nest-box within each hour, and the relative proportion of observations that were normothermic or hyperthermic for each individual. Our definition of hyperthermia allows for individual variation in the HI (and consequently the upper bound on normothermia), and thus controls for individual differences in resting $T_b$.

### 2.3.3. Ambient temperature data

Hourly temperature readings were acquired from Trent University's weather station (Environment Canada, http://climate.weather.gc.ca/index_e.html), which is located approximately 1.5 km from the Sewage Lagoon and approximately 9.5 km from the Trent Nature Areas. For all analyses, we excluded data in which the ambient temperature was greater than 30.9°C, because we did not have $T_b$ and feeding rate measurements above this value for control birds, and because all values exceeding 30.9°C were attributed to one late-nesting individual in the trimmed treatment. We, therefore, chose to keep

our interpretations conservative, and not extrapolate our predictions outside of the data available for both treatments. The average daily temperature ± s.d. (days 3–14 post-hatch and 05.00–21.00 h inclusive) for our dataset was 20.9 ± 5.3°C (range, 6.4–30.9°C).

## 2.4. Statistical analyses

For all analyses, we checked that our models met assumptions of normality, homogeneity of variance and independence. We considered $p$-values ≤ 0.05 as statistically significant. We calculated confidence intervals with the Wald method, in accordance with the R-package used for each analysis (stated in text).

### 2.4.1. Testing for anticipatory regulation using $T_b$ and feeding rate

One of our primary study goals was to test the hypothesis that chick-rearing birds regulate $T_b$ in an anticipatory fashion. A direct test of this hypothesis would require continuous readings of $T_b$ and feeding rate prior to and during flight, and a subsequent fine-scale analysis focusing on the effects of $T_b$ on feeding rate (*sensu* [40]). Because we did not have $T_b$ and activity level of individuals when away from the nest-box, we tested anticipatory regulation indirectly by examining how treatment affected $T_b$ and feeding rate in two separate linear mixed effects models (nlme package, [41]).

In the first model, we assigned $T_b$ as the response variable (Gaussian distributed), and included treatment, hourly $T_a$, hourly feeding rate (per chick), hour (i.e. time of day), nestling age, maternal age and interaction terms for treatment × $T_a$ and treatment × feeding rate as predictors. The interaction terms allowed us to assess how $T_b$ varied with treatment across a gradient of ambient temperatures, and across different levels of feeding rates. Brood size corresponded to the number of chicks in the nest on the day of feeding rate measurement. For brood size on day 14, we used day 12 brood size (because we did not visit nests after day 12). To account for nonlinearity between $T_b$ and $T_a$, we modelled $T_a$, as well as the 'treatment × $T_a$' interaction term, with a third-order polynomial. We determined the degree of nonlinearity between $T_b$ and $T_a$ in our model by visual examination of the raw data, and by running a likelihood-ratio test based on the log-likelihoods (which also produced Akaike's information criterion (AIC) values calculated from three different models (estimated with maximum likelihood): $T_a$ as a linear term ($T_a^1$) (model 1), $T_a$ as a quadratic term ($T_a^2$) (model 2) and $T_a$ as a cubic term ($T_a^3$) (model 3). Visual inspection of the raw data suggested a cubic relationship between $T_b$ and $T_a$, which was supported by the significantly higher log-likelihood in the cubic term model than the other two models (Log $\mathcal{L}$ Model 1: −81.84, AIC = 187.69, d.f. = 12; Log $\mathcal{L}$ Model 2 = −70.59, AIC = −169.18, d.f. = 14, $\mathcal{L}$ Ratio = 22.51, $p <$ 0.001; Log $\mathcal{L}$ Model 3 = −65.44, AIC = 162.90, d.f. = 16, $\mathcal{L}$ Ratio = 10.28, $p = 0.006$). Additionally, we weighted observations by the inverse variance of feeding rate, as we detected heteroskedasticity (decreasing variance in feeding rate) in the feeding rate residuals.

### 2.4.2. Testing for occurrence of normothermia and hyperthermia

To determine if tree swallows employed facultative hyperthermia during nestling provisioning, we first tested whether the hyperthermia threshold (i.e. mean modal $T_b$ ± HI, or the upper bound on $T_b$ during 'resting') differed between treatments using a two-tailed independent sample $t$-test. We then determined how treatment affected the relative frequency of hyperthermia by running two separate generalized linear mixed models (GLMM) (glmmTMB package, [42]). We initially attempted to model the frequency of hyperthermia in one analysis, but due to a high degree of zero-inflation (i.e. there were many observations in which birds were not hyperthermic), the model had poor predictive power, even with the addition of a zero-inflation term. We, therefore, took a two-step approach, first partitioning our data binomially, based on whether birds did (1) or did not (0) experience hyperthermia within each hour. In the second model, we took only the hours in which we observed an instance of hyperthermia (approx. 45% of all hours), and subsequently determined the proportion of observations that birds were hyperthermic versus normothermic (i.e. number of hyperthymic observations/number of total observations) within each hour, approximated with a Tweedie error distribution (to capture the Poisson-like distribution in our continuous data). Our fixed predictors in each model were treatment, $T_a$, hourly feeding rate (per chick), hour (i.e. time of day), maternal age, the HI threshold (i.e. modal $T_b$ + HI) for each bird and a treatment × $T_a$ interaction. We included the HI threshold as a covariate in our analysis because there was a large amount of individual variation in the estimated upper bound on normothermia (i.e. modal $T_b$ + HI, range = 41.3 – 44.7°C, s.d. = 1.09°C). We initially included individual identity as a random effect and a first-order autocorrelation structure in both models to control for

repeated measures across hours from the same individual. In the first model (i.e. probability of hyperthermia); however, residual plots suggested no evidence of autocorrelation, and so we did not include the autocorrelation term in the final model. In the second model, we kept both the random intercept and autocorrelation structure. In the first model (entire dataset), the sample size (i.e. number of birds) in each group were $n_{control} = 8$ and $n_{trimmed} = 8$ and in the second model, $n_{control} = 6$ and $n_{trimmed} = 7$.

# 3. Results

## 3.1. Body temperature is driven largely by anticipatory regulation

The mean $T_b$ (± 1 s.e.m.) for control and trimmed birds averaged across all days was 42.0 ± 0.20°C and 41.7 ± 0.19°C, respectively. $T_b$ varied with $T_a$ in a nonlinear fashion (i.e. third-order polynomial, $T_a^3$, table 1), and increased with feeding rate, nestling age and hour of the day, but was unrelated to maternal age (table 1). While the relationship between $T_b$ and $T_a$ was nonlinear, it differed as a function of treatment, with trimmed birds maintaining lower $T_b$ than controls across most of the $T_a$ range (except at low $T_a$; i.e. Treatment × $T_a^2$, table 1 and figure 1a). The relationship between $T_b$ and feeding rate also differed as a function of treatment, with trimmed birds maintaining lower $T_b$ than controls, given the same $T_a$ and same feeding rate (i.e. Treatment × Feeding Rate, table 1 and figure 1b). This indicates that for trimmed individuals, the extra capacity to dissipate heat allowed individuals to provision at higher rates without incurring the cost of a higher $T_b$ (see electronic supplementary material, figure S1). Although $T_b$ rose with feeding rate in control birds, $T_b$ primarily remained below the hyperthermia threshold (see figure 1a,b), consistent with anticipatory regulation. The interactive effect of treatment and ambient temperature on maternal feeding rate has been reported elsewhere [29] but even with the subset of data used here (reported in electronic supplementary material), trimmed birds maintained higher feeding rates than controls at high $T_a$, thereby supporting anticipatory regulation.

## 3.2. Individuals exhibit facultative hyperthermia only at the hottest temperatures

The mean resting modal $T_b$ ± HI (i.e. normothermia) was 41.19 ± 1.61°C (upper bound = 42.8°C) and 41.38 ± 1.62°C (upper bound = 43.0°C) for control and trimmed birds, respectively. The hyperthermia threshold (i.e. upper bound on resting modal $T_b$) did not, however, differ between treatments ($t = -0.37$, d.f. = 13.19, 95% CI [−1.23, 0.92], $p = 0.719$, figure 1a,b). The probability that an individual would exhibit at least one instance of hyperthermia in any given hour did not differ between control and trimmed individuals (table 2A; estimated marginal means ± 95% CI, Control: 50.8 ± [13.2, 87.5]%; Trimmed: 10.3 ± [0.02, 42.6]%), although this may be a function of small sample size. This is consistent with anticipatory regulation, because we predicted that if hyperthermia were a mechanism to maintain feeding rates, control birds would become hyperthermic more frequently than trimmed birds (but they did not). Further, while the probability of hyperthermia increased with $T_a$ for both groups (table 2A), the shape of the relationship between probability of hyperthermia and $T_a$ did not differ between treatments, again indicating that controls were no more likely to experience hyperthermia, even at high $T_a$ (i.e. Treatment × $T_a$, table 2A). In fact, in the control group, only 14.9% of all observations (i.e. all body temperature measurements) were hyperthermic (3992/26 612), which fell to 9.4% (3793/40 179) in the trimmed group. The probability that an individual would become hyperthermic increased with feeding rate, independent of experimental treatment, suggesting that harder working birds may employ facultative hyperthermia more frequently than birds that worked less (i.e. Feeding rate, table 2A). The probability of hyperthermia decreased with nestling age (table 2A), and with the HI cut-off (table 2A), indicating that individuals were less likely to become hyperthermic as their chicks grew.

During hours in which at least one observation of hyperthermia was observed, the frequency with which individuals became hyperthermic increased with $T_a$ (i.e. $T_a$, table 2B). On average, trimmed birds became hyperthermic less often than controls (i.e. Treatment, table 2B, figure 2), even while feeding at a higher frequency. For example, at 30°C, control birds made approximately 170 visits compared with the 240 visits of trimmed birds (given a 16 h day, and six chicks), a difference of 28% (see electronic supplementary material). At 30°C, control birds were hyperthermic 43% of the time (estimated marginal mean) compared with only 17% of the time (estimated marginal mean) in trimmed birds (figure 2), suggesting that control birds employed facultative hyperthermia with increasing frequency at high temperatures when compared with trimmed birds. The relationship between $T_a$ and the frequency of hyperthermic events did not differ

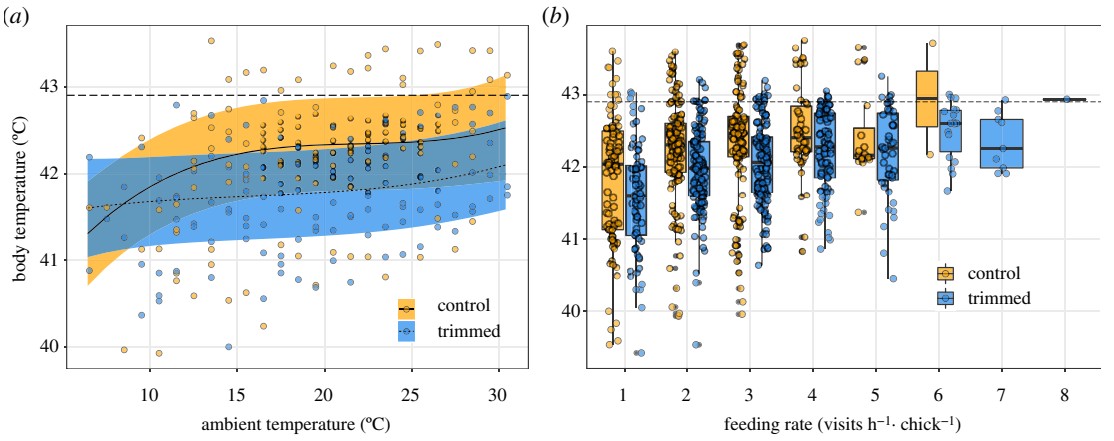

**Figure 1.** (a) The effect of treatment (feather trimming) on the relationship between body temperature and ambient temperature in female tree swallows, as determined via a linear mixed effects model (LMM). Trend lines and bands represent predicted marginal means ± 95% confidence intervals. Dots represent the raw $T_b$ averaged per bird at each temperature. The shape of the relationship between $T_b$ and $T_a$ differed between treatments (treatment $\times T_a^2$, $p = 0.006$). (b) The relationship between body temperature and feeding rate, as shown with raw data (boxplots and points). For visual purposes, feeding rate (visits $h^{-1}$ chick$^{-1}$) has been rounded up to the nearest whole number. Box limits represent the interquartile range, lines represent the median and whiskers represent $\pm 1.58 \times$ interquartile range. The slope of the relationship between $T_b$ and feeding rate differed between treatments (i.e. treatment $\times$ feeding rate, $p = 0.028$). (a,b) Long-dashed horizontal line across the top of the plots represents the hyperthermia cut-off (i.e. the modal $T_b$ + HI), averaged for both treatments. Above this line, individuals were predicted to be hyperthermic.

**Table 1.** Factors predicting $T_b$ in tree swallows with (i.e. trimmed) and without (i.e. control) an experimental capacity to dissipate heat. Square brackets next to terms indicate the reference category. Italicized p-values are statistically significant at less than or equal to 0.05 threshold. Superscripts on $T_a$ indicate order of polynomial.

| predictor | estimate | 95% CI | t-value | p-value |
|---|---|---|---|---|
| intercept [control] | 41.71 | 41.20, 42.22 | 161.10 | *<0.001* |
| treatment [trimmed] | −0.38 | −1.12, 0.36 | −1.11 | 0.287 |
| $T_a^1$ | 3.35 | 1.74, 4.96 | 4.09 | *<0.001* |
| $T_a^2$ | −1.50 | −2.95, −0.04 | −2.01 | *0.044* |
| $T_a^3$ | 1.49 | 0.50, 2.49 | 2.94 | *0.003* |
| feeding rate | 0.05 | 0.02, 0.08 | 3.58 | *<0.001* |
| nestling age | 0.03 | 0.01, 0.04 | 4.55 | *<0.001* |
| maternal age [second year] | 0.01 | −0.75, 0.76 | 0.02 | 0.984 |
| hour | 0.02 | 0.01, 0.02 | 4.98 | *<0.001* |
| treatment $\times T_a^1$ | −0.77 | −2.70, 1.15 | −0.79 | 0.430 |
| treatment $\times T_a^2$ | 2.34 | 0.68, 4.01 | 2.77 | *0.006* |
| treatment $\times T_a^3$ | −1.06 | −2.27, 0.15 | −1.72 | 0.086 |
| treatment $\times$ feeding rate | −0.04 | −0.07, 0.00 | −2.20 | *0.028* |

with respect to treatment (i.e. Treatment $\times T_a$, table 2B), indicating that both groups employed facultative hyperthermia to some degree as temperatures increased. The number of hyperthermic events increased with nestling age (table 2B) but did not differ with respect to feeding rate (table 2B), suggesting that once an individual is hyperthermic, $T_a$ appears to be the primary determinant of hyperthermia.

## 4. Discussion

Our results provide evidence that exercising birds regulate $T_b$ with a combination of anticipatory regulation and facultative hyperthermia and suggest that anticipatory regulation is the more dominant

**Table 2.** Factors predicting (A) the probability of hyperthermia and (B) the proportion of hyperthermic observations in tree swallows with (i.e. trimmed) and without (i.e. control) an experimental capacity to dissipate heat. Note that the analysis for 2B only includes hours where hyperthermia was observed. Estimates are presented on the linear scale. Square brackets next to terms indicate the reference category. Italicized $p$-values are statistically significant at less than or equal to 0.05 threshold.

| A | | | | |
| --- | --- | --- | --- | --- |
| predictor | estimate (log-odds) | 95% CI | $z$-value | $p$-value |
| intercept [control] | 178.22 | 109.51, 246.92 | 5.08 | *<0.001* |
| treatment [trimmed] | 0.41 | −3.53, 4.34 | 0.20 | 0.839 |
| $T_a$ | 0.23 | 0.09, 0.37 | 3.16 | *0.002* |
| feeding rate | 0.36 | 0.16, 0.56 | 3.53 | *<0.001* |
| nestling age | −0.17 | −0.25, −0.08 | −3.80 | *<0.001* |
| maternal age [second year] | 0.30 | −2.31, 2.92 | 0.23 | 0.822 |
| hour | 0.02 | −0.03, 0.08 | 0.82 | 0.411 |
| HI cut-off | −4.24 | −5.85, −2.63 | −5.15 | *<0.001* |
| treatment $\times T_a$ | −0.12 | −0.28, 0.03 | −1.59 | 0.112 |
| B | | | | |
| predictor | estimate | 95% CI | $z$-value | $p$-value |
| intercept [control] | 38.05 | 18.55, 57.55 | 3.83 | *<0.001* |
| treatment [trimmed] | −1.85 | −2.97, −0.74 | −3.25 | *0.001* |
| $T_a$ | 0.04 | 0.01, 0.07 | 2.60 | *0.009* |
| feeding rate | −0.04 | −0.01, 0.04 | −1.31 | 0.190 |
| nestling age | 0.08 | 0.02, 0.13 | 2.86 | *0.004* |
| maternal age [second year] | −0.52 | −0.09, 0.02 | −1.48 | 0.139 |
| hour | 0.01 | −1.21, 0.17 | 1.20 | 0.232 |
| HI cut-off | −0.95 | −1.41, −0.49 | −4.06 | *<0.001* |
| treatment $\times T_a$ | 0.03 | −0.01, 0.07 | 1.40 | 0.163 |

strategy. We found that control birds had higher $T_b$ and spent a greater proportion of time hyperthermic than trimmed birds at high $T_a$ (figure 1a, figure 2, respectively), indicating that tree swallows employ facultative hyperthermia when it is hot. However, control birds decreased feeding rates with increasing $T_a$, while trimmed birds maintained a constant feeding rate with increasing $T_a$, indicating that control individuals adjusted workload to mitigate rising $T_b$, consistent with anticipatory regulation.

## 4.1. Modulating feeding rate as a mechanism to regulate $T_b$

In birds, and endotherms in general, $T_b$ rises with $T_a$ during exercise [18,43,44]. We, therefore, predicted that in provisioning swallows, $T_b$ would rise with $T_a$, but that if birds use anticipatory regulation to regulate $T_b$, control and trimmed birds would have similar average $T_b$, even at higher $T_a$. In control birds, we found that with increasing $T_a$ (i.e. 10–30°C), $T_b$ increased by approximately 0.7°C, which was reduced to approximately 0.4°C in trimmed birds (controlling for the effect of feeding rate on $T_b$) (figure 1a). This effect is small (a difference of 0.3°C), however, and neither controls nor trimmed birds surpassed the hyperthermia threshold frequently (of all observations, 14.9% for controls, 9.4% for trimmed birds), which is consistent with the strategy of anticipatory regulation, at least approximately 85% of the time. It is worth noting that in both treatments, $T_b$ appears as if it would have continued to increase more steeply above the maximum $T_a$ of 30.9°C (figure 1a). This suggests that the treatment-specific difference in $T_b$ would have been more pronounced at higher $T_a$, and possibly an increased use of facultative hyperthermia by control birds.

We predicted that if birds use facultative hyperthermia during exercise, both treatments would on average have similar feeding rates, even at higher $T_a$. Yet, previously (and again shown here in

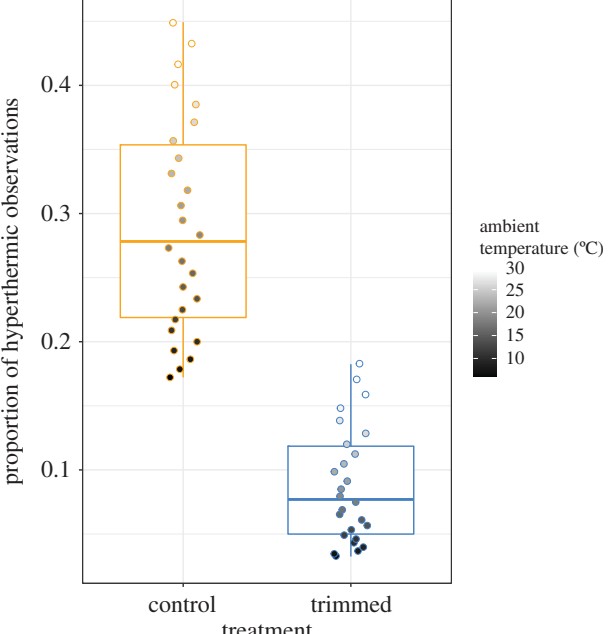

**Figure 2.** The relationship between the relative frequency of hyperthermia and $T_a$ as a function of heat dissipation capacity in female tree swallows. Box limits represent the interquartile range, lines represent the median and whiskers represent $\pm 1.58 \times$ interquartile range. Dots represent the predicted proportion of hyperthermic observations across a range of environmental temperatures (6–31°C). This model only includes hours in which at least one observation of hyperthermia was recorded. Trimmed birds experienced hyperthermia less often than control birds (i.e. Treatment, $p = 0.001$). Results were derived from a generalized linear mixed effects model.

electronic supplementary material), we found that controls had lower feeding rates than trimmed birds at high $T_a$ [29]. This indicates that control birds may have faced a heat dissipation limitation at high $T_a$, which is consistent with anticipatory regulation of $T_b$. This result is also consistent with human athletic performance research, in which performance tends to decline at higher $T_a$ [9,11,45], and cooling athletes either before or during competition improves performance [46]. It is also consistent with studies examining the effect of $T_a$ on activity level in other endotherms, which includes feeding behaviour in birds. For instance, both desert and non-desert birds have been found to decrease some aspect of their foraging effort (e.g. visit rate, food load, prey size) under high heat loads [47–51].

## 4.2. Hyperthermia occurs at highest ambient temperatures

In addition to examining the role that anticipatory regulation plays in avian $T_b$ regulation, we asked whether birds employed facultative hyperthermia as a potential avenue to enable offspring provisioning. We predicted that if provisioning birds employed facultative hyperthermia to maintain consistent provisioning rates across $T_a$, then control and trimmed birds would have similar provisioning rates, but trimmed birds would become hyperthermic less often than controls. While we found that provisioning rates were higher for trimmed compared with control birds at high $T_a$, our data provide evidence that control birds employed facultative hyperthermia. The probability (estimated marginal mean $\pm$ 95% CI) of control individuals experiencing at least one instance of hyperthermia in an hour was 89.4 $\pm$ [0.43, 0.99]% at 30°C during nestling provisioning. Further, when at least one instance of hyperthermia was observed in any given hour, we found that control birds spent proportionally more time hyperthermic within the hour than trimmed birds (43 versus 17%, at 30°C) (figure 2). That control birds were on average hyperthermic for 26 min h$^{-1}$ at highest $T_a$ indicates the use of facultative hyperthermia, at least when temperatures are high. If we had found no treatment-specific differences in either metric of hyperthermia, we would assume that on average individuals would solely be regulating $T_b$ with anticipatory regulation. Our results provide evidence that even temperate species may employ facultative hyperthermia to maintain activity levels, as has been established in desert birds [36,52,53]. Our results are also consistent with the idea that smaller species are more likely to use facultative hyperthermia than larger ones [53], which may be related to

a life-history trade-off between offspring provisioning and the physiological costs associated with hyperthermia [54–56], but could also be for the purposes of water conservation [25]. Taken together, our findings show that although exercising tree swallows will sometimes employ facultative hyperthermia during chick-rearing, this seems to be a strategy reserved for the highest temperatures. At intermediate–lower temperatures, adjusting activity in an anticipatory way appears to be the dominant strategy to avoid dangerously high $T_b$.

## 4.3. Implications for life-history variation

Within species, individuals may adjust behaviour and/or physiology in response to differences in age, sex and/or condition. For instance, experimental facilitation of heat loss in breeding blue tits (*Cyanistes caeruleus*) reduced average $T_b$, but only young females (i.e. those lacking breeding experience) produced heavier chicks [28]. The authors suggested that younger birds must work harder than older ones to achieve equivalent breeding success, and as a result, the capacity to dissipate heat generated through activity may have greater reproductive consequences for young individuals.

Strategies for regulating body temperature may also be condition-dependent. For example, robust immune or oxidative defences may aid in dealing with costs of hyperthermia (e.g. increased oxidative stress, [54–56]). When blue tits were given an experimentally enhanced thermal window, individuals had increased innate immune defences when compared with controls, suggesting the release of a constraint on heat dissipation allowed for increased allocation towards self-maintenance [57]. While we did not examine interactions among age, immune-status and work rate on $T_b$, these effects may still be important in our population and may contribute to variation in reproductive success.

Across species, susceptibility to hyperthermia may lead to population differences in life-history strategies. In tree swallows, for instance, clutch size increases with latitude, with northern populations raising more offspring per season than southern populations [58]. While speculative, such differences are consistent with the hypothesis that an individual's sustained energy expenditure is limited by its capacity to dissipate metabolically generated heat (heat dissipation limitation hypothesis, [59]).

## 5. Conclusion

We sought to understand what strategies breeding birds use to manage their body temperature during the metabolically demanding period of nestling provisioning. We tested two non-mutually exclusive hypotheses, i.e. anticipatory regulation and facultative hyperthermia, to understand how tree swallows manage the risks of overheating. Our results support the use of both anticipatory regulation and facultative hyperthermia; tree swallows mitigated the risk of hyperthermia by reducing workload at high $T_a$ but allowed $T_b$ to rise above normothermic levels at high $T_a$. When individuals were provided with an increased capacity for heat dissipation, via feather trimming, they became hyperthermic less frequently, which enabled them to maintain a higher workload at high $T_a$.

To our knowledge, this study provides the first evidence of a temperate breeding bird employing facultative hyperthermia during offspring provisioning. Previous authors have suggested that regulated hyperthermia may occur more frequently in species or populations inhabiting warmer versus cooler climates [53]. Tree swallows are a good model to test this prediction, because there are populations inhabiting both warm and cool climates, which show differences in their behavioural foraging and life-history strategies [60].

Our study has implications in the context of global warming. The maximum ambient temperature experienced by the birds in our study was approximately 31°C, which is lower than temperatures experienced by birds in more tropical and arid climates, and yet we saw a decline in provisioning performance at that temperature. As extreme climatic events are predicted to increase in frequency and severity [61], it remains to be seen whether there will be concomitant widespread decreases in provisioning performance for tree swallows and other species. It is possible that under differing environmental conditions, the degree to which individuals may choose to employ facultative hyperthermia may differ. It is, therefore, important, as a next step, to assess what the physiological costs may be for free-ranging animals.

Ethics. All research was approved by the Trent University Animal Care Committee, in accordance with the Canadian Council on Animal Care (AUP no. 24747).

Data accessibility. Data and code are available at the Dryad Digital Repository: https://doi.org/10.5061/dryad.r2280gbb4 [62].

Authors' contributions. S.T., J.J.N. and G.B. conceptualized the study; S.T. carried out data collection, formal analyses and wrote the original draft of the manuscript. S.T., J.J.N. and G.B. critically revised the manuscript.

Competing interests. The authors declare no competing or financial interests.

Funding. This research was supported by funds from the Natural Sciences and Engineering Research Council (NSERC) (RGPIN-04158-2014) and internal research grant provided by Trent University. S.T. was supported in part by an Ontario Graduate Scholarship.

Acknowledgements. We thank several people for their contribution to this manuscript, including Justin Boyles who provided advice on quantifying hyperthermia, and Josh Robertson, Samantha Morin and members of the Burness Lab for advice on statistics and feedback on the manuscript. We thank Alexander Gerson for loaning us PIT tag equipment, and Aleesa Schubert, Aaron Dain and Jenn Baici for help with data collection. We also thank several editors and reviewers for their helpful comments on previous versions of this manuscript.

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
