## [Reviewer comments · Royal Society Open Science]

Review History

Decision letter (RSOS-201589.R0)

Dear Mr Tapper:

It is a pleasure to accept your manuscript entitled "**Experimental evidence that hyperthermia limits offspring provisioning in a temperate-breeding bird**" in its current form for publication in Royal Society Open Science. The comments of the reviewer(s) who reviewed your manuscript are included at the foot of this letter.

Please ensure you make the minor typographical adjustments during the proofing process.

Reports © 2020 The Reviewers; Decision Letters © 2020 The Reviewers and Editors; Responses © 2020 The Reviewers, Editors and Authors. Published by the Royal Society under the terms of the Creative Commons Attribution License <http://creativecommons.org/licenses/by/4.0/>, which permits unrestricted use, provided the original author and source are credited

on behalf of Dr Cynthia Downs (Associate Editor) and Professor Kevin Padian (Subject Editor).

Associate Editor Dr Cynthia Downs Comments to Author:

Associate Editor

Comments to the Author:

This manuscript was previously reviewed by three reviewers and an editor at Proc B and was transferred to Open Science. The version transferred to Open Science included revisions to address the previous reviews. The study presented demonstrated that environmental temperatures mediate behavior and that hyperthermia can limit offspring provisioning. This straightforward, experimental test of this concept is essential to document in the literature, although it is not the first test of this concept. The authors' revisions to the manuscript sufficiently address the reviewers' comments, the experiment is scientifically sound, and the manuscript is clearly written.

Line edits:

Line 32: Add "treatment" before individuals to clarify that treatment individuals are still being discussed.

Line 135: Change "between" to "among"
